# Identification of Circulating Serum miRNAs as Novel Biomarkers in Pancreatic Cancer Using a Penalized Algorithm

**DOI:** 10.3390/ijms22031007

**Published:** 2021-01-20

**Authors:** Jaehoon Lee, Hee Seung Lee, Soo Been Park, Chanyang Kim, Kahee Kim, Dawoon E. Jung, Si Young Song

**Affiliations:** 1Department of Statistics, Seoul National University, Seoul 08733, Korea; jhlee1213@gmail.com; 2Division of Gastroenterology, Department of Internal Medicine, Yonsei University College of Medicine, Seoul 03722, Korea; LHS6865@yuhs.ac (H.S.L.); ELLIE0813@yuhs.ac (S.B.P.); MISSKCY90@yuhs.ac (C.K.); KIMMY2378@gmail.com (K.K.); 3Institute of Gastroenterology, Department of Internal Medicine, Yonsei University College of Medicine, Seoul 03722, Korea

**Keywords:** microRNA, pancreatic cancer, diagnosis, sensitivity, specificity, biomarker

## Abstract

Pancreatic cancer (PC) is difficult to detect in the early stages; thus, identifying specific and sensitive biomarkers for PC diagnosis is crucial, especially in the case of early-stage tumors. Circulating microRNAs are promising non-invasive biomarkers. Therefore, we aimed to identify non-invasive miRNA biomarkers and build a model for PC diagnosis. For the training model, blood serum samples from 63 PC patients and 63 control subjects were used. We selected 39 miRNA markers using a smoothly clipped absolute deviation-based penalized support vector machine and built a PC diagnosis model. From the double cross-validation, the average test AUC was 0.98. We validated the diagnosis model using independent samples from 25 PC patients and 81 patients with intrahepatic cholangiocarcinoma (ICC) and compared the results with those obtained from the diagnosis using carbohydrate antigen 19-9. For the markers miR-155-5p, miR-4284, miR-346, miR-7145-5p, miR-5100, miR-661, miR-22-3p, miR-4486, let-7b-5p, and miR-4703-5p, we conducted quantitative reverse transcription PCR using samples from 17 independent PC patients, 8 ICC patients, and 8 healthy individuals. Differential expression was observed in samples from PC patients. The diagnosis model based on the identified markers showed high sensitivity and specificity for PC detection and is potentially useful for early PC diagnosis.

## 1. Introduction

Pancreatic cancer (PC) is one of the leading causes of cancer-related mortality, as the symptoms of PC seldom appear in the early stages of the disease, and the cancer is mostly detected after it has metastasized to other organs. According to cancer statistics in 2020, the five-year survival rate of patients with PC is 9%, although that of patients with localized PC is higher than 37%, based on patients diagnosed with pancreatic cancer between 2009 and 2015 [1].

The most effective strategy for reducing PC-related mortality is early diagnosis and treatment. However, the lack of reliable markers for PC detection reduces the efficacy of screening strategies in at-risk populations, such as those with chronic pancreatitis [2]. Carbohydrate antigen 19-9 (CA19-9) and carcinoembryonic antigen (CEA) are the most commonly used serological biomarkers; however, they lack sufficient sensitivity and specificity for the detection of PC [3]. To improve the prognosis of patients with this form of cancer, it is important to identify diagnostic biomarkers for PC.

Recently, microRNAs (miRNAs), which are small non-coding RNA molecules, have been reported to play important roles in post-transcriptional regulation in cancer [4]. Increasing evidence has shown that miRNAs are essential for the development, diagnosis, and prognosis of cancer, suggesting that these RNAs have potential for use as diagnostic markers in cancer [5]. To date, nearly 100 miRNAs have been identified to be associated with PC using tissue samples [2]. However, it is difficult to perform tissue biopsies in every patient suspected of having PC. Therefore, the optimal biomarkers would be non-invasive and derived from blood, such as circulating miRNAs, which may be readily collected from the patient. Another reason for using circulating miRNAs as biomarkers is their remarkable stability in plasma and serum. They are protected from RNAse degradation as they can be packaged in microparticles (e.g., exosomes) or bound to Argonaut proteins or high-density lipoproteins [6,7,8,9,10].

Currently, highly sensitive and specific invasive biomarkers are not available for the detection of PC. Therefore, the primary objective of this study was to identify non-invasive miRNA biomarkers and to build a prediction model for the diagnosis of PC. In this study, 63 PC patients and 63 control subjects were used for the identification of miRNA biomarkers, and an additional 25 PC samples and 81 intrahepatic cholangiocarcinoma (ICC) samples were used for the validation of our proposed prediction model. For comparison, we also obtained diagnosis results based on serum levels of CA19-9 in the same blood samples. For additional validation, quantitative reverse transcription PCR (qRT-PCR) was conducted using additional RNA samples from 17 patients with PC, 8 patients with ICC, and 8 healthy individuals.

## 2. Materials and Methods

### 2.1. Study Design

The present study included 105 patients with PC, 109 patients with ICC, 7 patients with stomach cancer (SC), 5 patients with colorectal cancer (CRC), 2 patients with gastrointestinal stromal tumor (GIST), 10 patients with cholelithiasis (Ch), and 27 healthy subjects who had been clinically classified at the time of participation. A case-control study was designed to identify differentially expressed miRNAs (DEmiRNAs) between the case-control groups and to build a diagnostic model for PC. For cases, 63 PC patients were used, and for controls, two types of control groups were used. 

The first type of control group consisted of 19 healthy subjects and 10 Ch patients. The second type of control group, the non-PC group, included samples from patients with other cancers as well as those from non-cancer subjects. In particular, we included 20 ICC patients, 7 SC patients, 5 CRC patients, and 2 GIST patients. We set aside 25 PC and 81 ICC samples for the validation study. The clinical characteristics of the samples in the microarray experiments and the grouping details are presented in Table 1. qRT-PCR was conducted using samples from 17 PC patients, 8 ICC patients, and 8 healthy individuals. The purpose of our study was not only to identify biomarkers for PC but also to build a prediction model. Therefore, although the age of the subjects was significantly different between the case and control groups, we decided to use the model without the covariate, as the model with the covariate had a similar prediction performance to the model without the covariate. The study protocol conformed to the ethical guidelines of the 1975 Declaration of Helsinki, and the Ethical Committee and Institutional Review Board of Yonsei University College of Medicine approved the protocol of serum acquisition from the patients’ specimens. Written informed consent was obtained from all participating patients and healthy controls (IRB approval code 4-2012-0528, 20 September 2012).

### 2.2. Sample Preparation

Patient samples were prospectively obtained from consenting individuals who underwent a detailed clinical examination and were diagnosed at the Severance Hospital, Yonsei University College of Medicine. Serum samples from 63 patients with PC, 63 non-PC control subjects, and another 25 patients with PC and 81 patients with ICC were collected in 10-mL BD serum tubes. Samples were centrifuged at 4 °C for 20 min at 3000× *g*. The supernatant serum was then aliquoted and stored at −80 °C until further use.

### 2.3. MicroRNA Extraction

Total RNA containing miRNA was extracted from the serum samples using a serum miRNA purification kit (Genolution, Seoul, Korea) according to the manufacturer’s instructions, and the RNA was resuspended in 12 μL of RNase-free water and stored at −80 °C until microarray or qRT-PCR analysis.

### 2.4. MicroRNA Microarray Experiments

For quality control, the purity and integrity of the RNA were evaluated based on the OD260/280 ratio and analyzed using the Agilent 2100 Bioanalyzer (Agilent Technologies, Palo Alto, CA, USA). Analysis using the Affymetrix GeneChip miRNA 4.0 array (Affymetrix, Santa Clara, CA, USA) was performed according to the manufacturer’s protocol. RNA samples (130 ng) were labeled using the FlashTag Biotin RNA Labeling Kit (Genisphere, Hatfield, PA, USA). The labeled RNA was quantified, fractionated, and hybridized to the miRNA microarray according to the standard procedures provided by the manufacturer.

Next, the labeled RNA was heated to 99 °C for 5 min and then to 45 °C for 5 min. RNA-array hybridization was performed with agitation at 60 rotations per minute for 16 h at 48 °C on an Affymetrix 450 Fluidics Station. The chips were washed and stained using a GeneChip Fluidics Station 450 (Affymetrix). The chips were then scanned using an Affymetrix GCS 3000 scanner; 232 CEL files were analyzed and normalized using the Expression Console software. The Affymetrix GeneChip Micro 4.0 Array provides 100% miRBase v20 coverage (www.mirbase.org) using a one-color approach. This chip contains 6658 human probe sets, which includes pre-mature miRNAs (*n* = 2025) and other small RNAs (*n* = 1996), including internal and negative controls. For further analysis, we extracted 2578 mature human miRNAs, from all probe sets.

### 2.5. Principal Component Analysis Based on Differentially Expressed Genes

Log2-transformed and normalized intensities for the 2578 human mature miRNAs were analyzed for the difference in expression levels between the cases and controls. To identify DEmiRNAs, we used a logistic regression analysis. Statistical significance was determined using the false discovery rate (FDR) method; FDR < 0.05 was considered significant in this analysis.

To examine the difference in miRNA profiles between the cases and controls, we conducted a principal component analysis (PCA). The principal components of the two groups were computed based on different sets of miRNAs: (i) all miRNAs and (ii) FDR < 0.05. Based on this PCA model, we also predicted the principal components of the validation samples (25 PC samples and 81 ICC samples). To visualize the pattern of each group, we added 95% confidence ellipses of principal components in a PCA plot based on the multivariate *t* distribution.

### 2.6. Biomarker Selection for Diagnosis

For diagnosis of PC, miRNA biomarkers were selected from the 2578 human mature miRNAs, using the following procedure:Step 1 (training/test data assigning):○Whole data were randomly divided into 5 approximately equal-sized subsets (folds).○Each of the five folds were considered test data, and the remaining folds were designated as training data (5-fold cross-validation).
Step 2 (candidate variable selection):○Using the individualized assigned training data, logistic regression analysis was conducted, and *p*-values and adjusted *p*-values (FDR) were computed for each miRNA.○First candidate miRNAs were selected (FDR < 0.05).○By applying a smoothly clipped absolute deviation (SCAD) penalty to the first candidate miRNAs, second candidate miRNAs with non-zero coefficients were selected.
Step 3 (repetition):○Steps 1 and 2 were repeated 200 times with random seed.○1000 (5-fold CV × 200 repetitions) sets of candidate miRNAs were obtained.
Step 4 (final variable selection by voting):○From the 1000 sets of candidate miRNAs, the frequency of each candidate miRNA was computed.○The candidate miRNAs were sorted by frequency.
Step 5 (prediction model building):○From the top *K* ranked miRNAs, the radial basis function (RBF)-kernel SVM model was built (*K* = 2, …, 50). ○From each model (*K* = 2, …, 50), the optimal hyperparameters and performance were calculated using double cross-validation. ○Optimal *K* was determined based on the performance of each model.○As a final model, RBF-kernel SVM with *K* top-ranked miRNAs was applied using the whole training dataset.


### 2.7. Double Cross-Validation

For the parametrization and validation of our diagnostic model, we used double cross-validation [11,12], which consists of inner and outer cross validation. We conducted the outer 5-fold cross validation to determine the optimal *K* and the inner 5-fold cross validation for the hyperparameter assignment of SVM. In the inner 5-fold cross validation, for the grid search of kernel hyperparameters, we assigned gamma values in the range of −2^−10^ to 2^10^ (−2^−10^, 2^−9^, …, 2^9^, 2^10^) and cost values in the range of −2^−7^ to 2^7^ (2^−7^, 2^−6^, …, 2^6^, 2^7^). In the outer 5-fold cross validation, the diagnostic models with *K* top-ranked miRNAs were applied to the test data, and the area under the curve (AUC), sensitivity, and specificity were calculated for each fold. We calculated these performances for several *K* values (*K* = 2, …, 50). This double cross-validation was repeated 20 times in random seeds. The performance metrics were then averaged for 5 folds and 20 repetitions. Based on this performance, we determined the final number of biomarkers (=*K*).

### 2.8. Smoothly Clipped Absolute Deviation (SCAD) Penalty

SCAD is a non-concave penalty function introduced by Fan and Li [13], and Zhang et al. [14] considered the sparse SVM with SCAD for feature selection. The SCAD-penalized term for each coefficient *t_j_* has the following form Equation (1) [14]:(1)$$p_{\lambda }\left(t_{j}\right)=\{\begin{array}{cc} \lambda \left|t_{j}\right| & if\;\left|t_{j}\right|<\lambda \\ -\frac{\left({\left|t_{j}\right|}^{2}-2a\lambda \left|t_{j}\right|+\lambda^{2}\right)}{2\left(a-1\right)} & if\;\lambda <\left|t_{j}\right|\leq a\lambda \\ \frac{\left(a+1\right)\lambda^{2}}{2} & if\;\left|t_{j}\right|>a\lambda \end{array}$$

In our analysis, Fan and Li’s suggested value for a = 3.7 was used. The parameter *λ* was assigned by minimizing the approximate generalized cross-validation statistics. Among the various penalized methods for feature selection, we chose SCAD because it has several desirable properties. For example, SCAD produces nearly unbiased estimates for large coefficients, and the set of features selected using SCAD are asymptotically equivalent to the set of true signal features; that is, SCAD satisfies the oracle property. We conducted a penalized SVM with the SCAD penalty for multiple miRNA selection in the double cross-validation in our study.

### 2.9. Quantitative RT-PCR

Reverse transcription and qRT-PCR were performed using a TaqMan Advanced miRNA cDNA Synthesis Kit (Applied Biosystems, Foster City, CA, USA), TaqMan Advanced miRNA Assays (Applied Biosystems), and TaqMan Fast Advanced Master Mix (Applied Biosystems), according to the manufacturer’s protocols. qRT-PCR was performed using an ABI Prism 7300 Sequence Detection System (Applied Biosystems), and primers for the mature miRNAs were purchased from Applied Biosystems. PCR amplification consisted of an initiation step at 95 °C for 10 min, followed by 55 cycles at 95 °C for 30 s, 56 °C for 30 s, and 72 °C for 15 s. All qRT-PCR assays were performed in triplicate using total RNA samples from 17 patients with PC, 8 patients with ICC, and 8 healthy individuals. Statistical analyses were analyzed using GraphPad 5 (GraphPad Software). The miRNA expression between groups were calculated by a one-way ANOVA and Bonferroni post-tests.

## 3. Results

### 3.1. Comparison between Case and Control by PCA 

Upon comparing the 63 PC samples with the 29 non-cancer samples in the DEmiRNA analysis, we identified 103 miRNAs that showed significant differences in expression between the two groups (FDR < 0.05) (Appendix A). When 103 miRNAs were used in the PCA, the 63 PC samples (green dots) and 29 non-cancer samples (red dots) were well-distinguished compared to when all miRNAs were used. Furthermore, 25 validation-PC samples (purple dots) had similar patterns to the 63 PC samples, as shown in Figure 1a,b. However, some of the 81 validation-ICC samples (blue points) had overlapping patterns with the PC case samples. Thus, if we used only non-cancer samples as controls, the biomarkers led to many false positives (for example, the biomarker could diagnose some ICC patients as PC patients) and were not appropriate for the PC-specific diagnostic model.

Upon comparing the 63 PC samples with the 63 non-PC samples, we identified 149 miRNAs that showed significant differential expression between the two groups (FDR < 0.05) (Appendix A). When we used all the miRNA data in the PCA, the PC and non-PC samples exhibited overlapping patterns of principal components (Figure 1c). When the 149 differentially expressed miRNAs were used in the PCA, the clustering patterns of the 63 PC samples and 63 non-PC samples were nearly distinguished, and the validation samples (25 PC and 81 ICC) had similar patterns to those of the training case-control samples, as shown in Figure 1d.

### 3.2. Building a Diagnostic Model Based on the Selected miRNA Markers

To build the diagnostic model, we decided to use the results of the comparison between the PC samples and non-PC samples, including samples from patients with other cancers as controls to obtain PC-specific diagnostic markers. For the selection of diagnostic markers, we used 5-fold cross validation with 200 repetitions. In each fold of the cross-validation, we conducted a logistic regression analysis without a covariate and selected a set of candidate miRNA markers whose FDR was less than 0.05. Then, through the use of the SVM with the SCAD penalty function, the candidate markers were narrowed down to the markers with non-zero coefficients.

We ranked the markers according to the selection frequency. Based on these frequencies, *K* top-ranked miRNAs were used to build the RBF kernel SVM model. To determine the value of *K*, through double cross-validation, we estimated the diagnostic performance of the model with the *K* top-ranked miRNAs by varying *K* (*K* = 1, …, 50). As shown in Figure 2, the performance measures increased as *K* increased and began to saturate at an AUC of 0.98 and an accuracy of 0.93 when *K* was 39. Therefore, we decided to select the top 39 miRNAs as diagnostic biomarkers for PC among the candidate miRNAs (Table 2).

At *K* = 39, the mean sensitivity and mean specificity of the diagnostic model were 0.93 and 0.93, respectively, given an optimal decision threshold. The optimal threshold of diagnosis probability was determined to be 0.55 by comparing the performance results based on thresholds (0.5, 0.55, 0.6, 0.65, and 0.7). Among the 39 miRNAs, 28 miRNAs were also differentially expressed between the PC samples and non-cancer samples (FDR < 0.05); 11 miRNAs were differentially expressed between the PC and non-PC samples (FDR < 0.05) (Figure 3).

For validation, we next applied our PC-specific diagnostic model to a different set of 25 PC and 81 ICC samples. When the PC-diagnosis probability from the diagnostic model was >0.55, we diagnosed the patient as having PC. We also applied CA19-9 diagnosis to the same samples for comparison. When the CA19-9 value was >37, we diagnosed the patient as having PC. As shown in Figure 4, the AUC of the proposed diagnostic model was 1.5 times higher, the sensitivity was 1.3 times higher, and the specificity was 2 times higher than that of the CA19-9 diagnosis model (the AUC, sensitivity, and specificity are presented in Figure 4).

We also validated 10 miRNAs of the 39 diagnostic markers using qRT-PCR. For qRT-PCR, blood samples from another 17 patients with PC, 8 patients with ICC, and 8 healthy individuals were used (Appendix A). The expression levels of miR-155-5p, miR-4284, let-346, miR-7154-5p, miR-5100, miR-661, miR-22-3p, miR-4486, let-7b-5p, and miR-4703-5p were analyzed using primers for mature miRNAs. The findings indicated differential expression in PC versus ICC and healthy individuals. Decreased expression of miR-155-5p, miR-7154-5p, miR-661, and miR-4703-5p and elevated expression of miR-5100, miR-22-3p, miR-4486, and let-7b-5p were observed in PC patients. miR-4284 was only detected in cancer groups and miR-346 was absent in patients with PC (Figure 5).

## 4. Discussion

Despite multiple clinical trials and continued efforts, PC remains the most difficult cancer to cure as it is difficult to diagnose at the early stages. In this study, we aimed to identify circulating miRNA biomarkers for the detection of PC and to develop a diagnostic model based on these markers. For the identification of diagnostic markers, we used two types of control group. The first control group consisted of 29 non-cancer samples and the second consisted of 63 non-PC samples, including those from patients with other cancers. DEmiRNAs selected from the PC vs. non-cancer study successfully enabled discrimination between the training-case samples and the training-control samples but could not distinguish the validation-case samples from the validation-control samples, possibly because the validation-control samples consisted of samples from patients with ICC. For validation control, patients with ICC were used instead of healthy patients to verify the specificity of PC diagnosis. PC and ICC are known to have overlapping immunohistochemical profiles [40]. DEmiRNAs selected from the PC vs. non-PC study differentiated the cases from the controls well, both in the training samples and in the validation samples. As a result, we found that the PC vs. non-PC grouping was more acceptable for identification of PC-specific diagnostic markers than the PC vs. non-cancer grouping. Based on this grouping, we tried to identify PC-specific diagnosis markers from 2578 miRNAs. In order to consider joint effects from multiple core miRNAs and filter the negative effects caused by irrelevant miRNAs, we conducted a penalized SVM with SCAD penalty. As a result, we identified 39 PC-specific diagnostic markers using SCAD-based penalized SVM with a double cross-validation technique.

Among the 39 PC-specific markers, 15 miRNAs have been reported to serve as PC markers in previous studies. Some of these (miR-155-5p, miR-939-5p, miR-346, miR-3064-5p, miR-661, miR-26a-5p, miR-200b-5p, and miR-218-5p) were found to be differentially expressed in PC tissues [15,16,17,18,19,20,23,24,27,30,31,32,34,35,37,38,39]. miR-939-5p, miR-200b-5p, and miR-218-5p were also differentially expressed in PC cell lines. miR-642b-3p and miR-22-3p, which have previously been reported as early diagnostic markers for PC, showed altered levels in the blood of patients with PC [21,22,33]. miR-4284 and miR-5100 are underexpressed in cyst fluid and saliva samples, respectively, in patients with PC [19,25,26]. miR-455-3p and miR-4745-5p have been found to be related to resistance to gemcitabine treatment in patients with PC [28,29]. Among the 39 markers obtained, several of them have also been reported to act as markers of other cancers, including colon, ovarian, breast, liver, lung, and prostate cancer. These markers, individually, are not specific to PC; however, we believe that our joint algorithm based on all 39 markers enables highly specific diagnosis of PC.

Among the present selected diagnostic markers, 10 markers were analyzed by qRT-PCR and differential expression was observed between cancer groups and healthy individuals. Previously reported PC markers including miR-155-5p, miR-4284, miR-5100, miR-346, miR-661, and miR-22-3p, as well as the novel markers including miR-7154-5p, miR-miR-4486, let-7b-5p, and miR-4703-5p, consistently showed differential expression in the PC samples, in both the microarray and qRT-PCR.

The markers identified in the present study have potential for use in the early diagnosis of PC and are expected to serve as a major platform for developing commercial models for the timely diagnosis of PC.

## 5. Conclusions

In this study, we identified 39 circulating miRNAs as PC-specific diagnostic markers using penalized methods. They include several novel biomarkers that have not yet been reported for PC diagnosis. For inner validation, we estimated the sensitivity and specificity of our diagnostic model through double cross-validation and obtained a mean sensitivity of 0.93 and mean specificity of 0.93. We also validated the specificity using 25 independent PC and 81 ICC samples with a PCA analysis and conducted qRT-PCR validation on several diagnostic markers using independent samples from 17 PC, 8 ICC, and 8 healthy control patients. qRT-PCR analysis indicated that miR-155-5p, miR-4284, miR-346, miR-7145-5p, miR-5100, miR-661, miR-22-3p, miR-4486, let-7b-5p, and miR-4703-5p were differentially expressed in samples from patients with PC. Overall, while we are convinced that our identified miRNA biomarkers based on the PC-specific diagnosis model improve the detection rate for PC, further validation studies will be needed in the future.

## Figures and Tables

**Figure 1 ijms-22-01007-f001:**
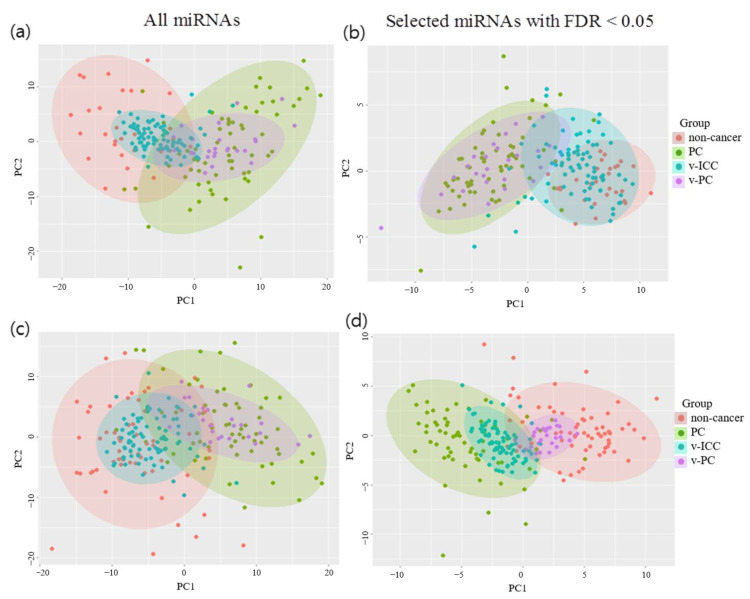
Principal component analysis by comparing case-control and validation samples based on different sets of miRNAs. Red, green, blue, and purple dots represent control, case, validation-control, and validation-case samples, respectively. (**a**,**b**) 63 pancreatic cancer (PC) cases and 29 non-cancer controls and (**c**,**d**) 63 PC cases and 63 non-PC controls. (**a**,**c**) All miRNAs and (**b**,**d**) selected miRNAs with false discovery rate <0.05. PC, pancreatic cancer; v-PC, validated-pancreatic cancer; FDR, false discovery rate; v-ICC, validated-intrahepatic cholangiocarcinoma.

**Figure 2 ijms-22-01007-f002:**
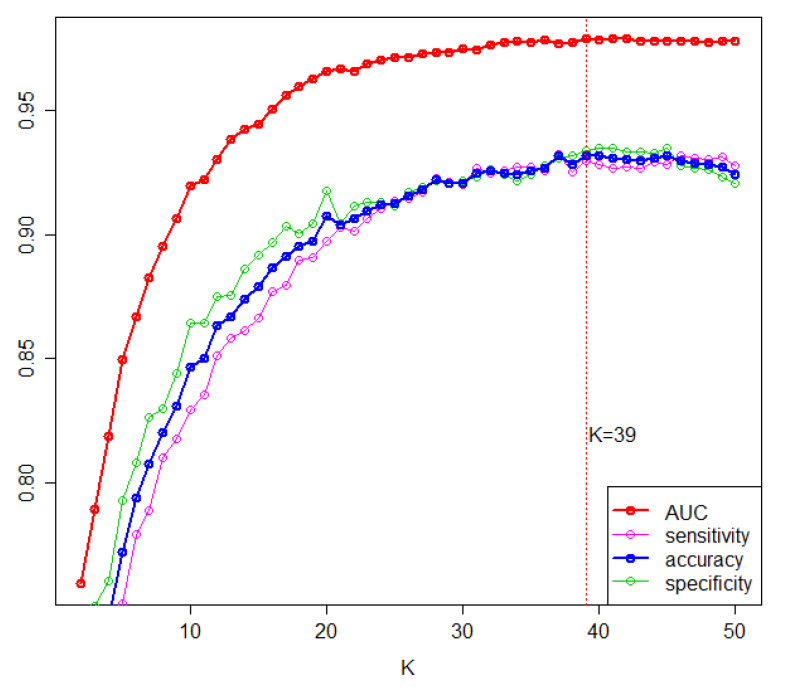
Predictive performance based on *K* top-ranked markers. From 200 repetitions of 5-fold cross validation, model prediction performance metrics, including area under the curve (AUC), sensitivity, and specificity, were averaged with specific *K*. Accuracy is the average of sensitivity and specificity.

**Figure 3 ijms-22-01007-f003:**
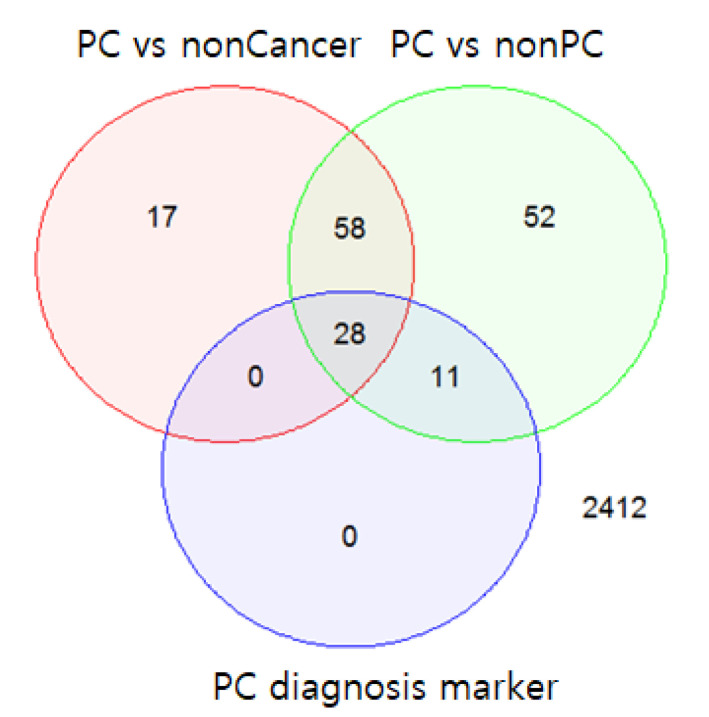
Venn diagram of the differentially expressed miRNAs and 39 selected diagnostic markers. The top 39 miRNAs included 28 miRNAs that were differentially expressed between pancreatic cancer (PC) samples and non-cancer samples (false discovery rate (FDR) < 0.05) and 11 miRNAs that were differentially expressed only between the PC and non-PC samples (FDR < 0.05).

**Figure 4 ijms-22-01007-f004:**
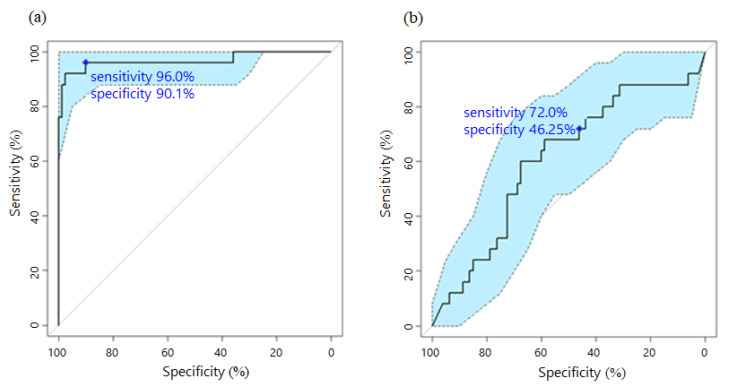
Receiver-operating characteristic (ROC) curve for pancreatic cancer (PC) diagnosis based on the proposed model and CA19-9. For validation, 25 PC samples and 81 intrahepatic cholangiocarcinoma samples were used to apply the (**a**) proposed model and (**b**) CA19-9 diagnosis. The confidence band of the ROC curve is indicated in light blue. The blue point represents the sensitivity and specificity based on the predefined threshold in each plot.

**Figure 5 ijms-22-01007-f005:**
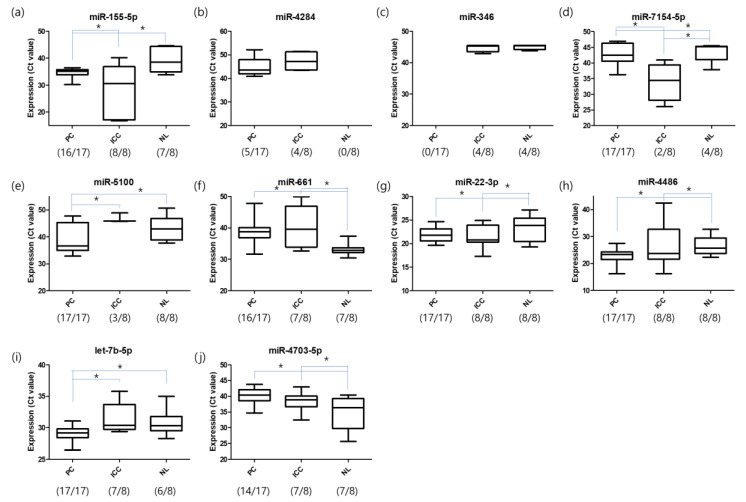
miRNA expression determined using quantitative reverse transcription (qRT-PCR). Differential expression levels of miRNAs selected from the chip data were further analyzed using qRT-PCR to confirm the expression levels in the serum samples of patients with PC (*n* = 17), patients with ICC (*n* = 8), and healthy individuals (*n* = 8). The expression levels of (**a**) miR-155-5p, (**b**) miR-4284, (**c**) let-346, (**d**) miR-7154-5p, (**e**) miR-5100, (**f**) miR-661, (**g**) miR-22-3p, (**h**) miR-4486, (**i**) let-7b-5p, and (**j**) miR-4703-5p were analyzed using primers for mature miRNAs (* *p*-value < 0.05). The number of samples with expression is indicated as (Number _Expression_/Number _Total_). PC, pancreatic cancer; ICC, intrahepatic cholangiocarcinoma; NL, normal or non-cancer.

**Table 1 ijms-22-01007-t001:** Clinical characteristic of the samples used for the identification and validation of the diagnostic markers.

Variables	Training Cohort							Validation Cohort	
	PC	ICC	CRC	SC	GIST	Ch	N	PC	ICC
Count	63	20	5	7	2	10	19	25	81
Age	63.0 ± 9.6	66.4 ± 10.5	66.2 ± 5.2	60.0 ± 14.4	54.5 ± 10.6	60.2 ± 12.1	46.8 ± 9.8	66.0 ± 8.1	64.6 ± 7.1
Female	19 (30.2)	8 (40.0)	2 (40.0)	1 (14.3)	1 (50.0)	6 (54.5)	6 (30.0)	7 (28.0)	6 (33.3)
Stage *									
I	4	1	1	1	2			1	7
II	12	7	2	1				4	18
III	13	-	-	1	-			8	8
IV	34	-	2	4	-			12	48
CA19-9, U/mL									
Median level	336.5 ± 6055.7	181.5 ± 4441.1	20.0 ± 28.8	62.1 ± 355.6	8.91 ± 9.89		10.1 ± 5.5	196.9 ± 5985.5	45.8 ± 5287.2
≤37	16 (25.4)	6 (30.0)	4 (80.0)	3 (42.9)	2 (100)		19 (100)	7 (28.0)	37 (46.8)
>37	47 (74.6)	14 (70.0)	1 (20.0)	4 (57.1)	0 (0)		0 (0)	18 (72.0)	42 (53.2)
Overall Survival, months	15.4	19.6	32.8	62.4	25.9		-	12	21.9

* Tumor stages were based on the staging classification of the 7th edition of the American Joint Committee on Cancer. Variables are expressed as mean ± standard deviation, median ± standard deviation, or n (%). PC, pancreatic cancer; ICC, intrahepatic cholangiocarcinoma; SC, stomach cancer; CRC, colorectal cancer; GIST, gastrointestinal stromal tumors; Ch, cholelithiasis; N, normal; SD, standard deviation; CA19-9, carbohydrate antigen 19-1.

**Table 2 ijms-22-01007-t002:** The identified 39 PC-specific diagnostic markers.

miRNA	Selection Freq	AUC	Reference
hsa-miR-548ay-5p *	950	0.734	
hsa-miR-155-5p *	842	0.808	[15,16,17,18]
hsa-miR-4284	836	0.708	[19]
hsa-miR-939-5p *	810	0.734	[20]
hsa-miR-642b-3p *	805	0.759	[21,22]
hsa-miR-346 *	736	0.749	[23]
hsa-miR-4690-5p *	690	0.716	
hsa-miR-7154-5p	675	0.698	
hsa-miR-3064-5p *	625	0.785	[24]
hsa-miR-1269b *	607	0.854	
hsa-miR-4708-3p *	580	0.82	
hsa-miR-5100 *	580	0.792	[25,26]
hsa-miR-548aq-3p *	580	0.76	
hsa-miR-661	523	0.701	[27]
hsa-miR-4701-3p *	509	0.699	
hsa-miR-1272 *	455	0.771	
hsa-miR-455-3p *	454	0.732	[28,29]
hsa-miR-26a-5p	422	0.711	[30,31,32]
hsa-miR-22-3p *	388	0.758	[21,33]
hsa-miR-6894-3p	384	0.697	
hsa-miR-3620-3p *	377	0.679	
hsa-miR-4775 *	377	0.759	
hsa-miR-4745-5p *	371	0.688	[29]
hsa-miR-6737-3p	371	0.673	
hsa-miR-5189-3p *	357	0.693	
hsa-miR-4647	356	0.741	
hsa-miR-4486 *	349	0.788	
hsa-miR-6865-5p *	345	0.764	
hsa-miR-200b-5p *	344	0.726	[34,35]
hsa-miR-548ac *	313	0.852	
hsa-let-7b-5p	298	0.676	
hsa-miR-2278 *	291	0.741	[36]
hsa-miR-4703-5p *	262	0.603	
hsa-miR-1226-5p	257	0.653	
hsa-miR-640 *	245	0.725	
hsa-miR-1277-3p	240	0.672	
hsa-miR-218-5p	232	0.624	[19,37,38,39]
hsa-miR-512-3p *	218	0.664	
hsa-miR-16-2-3p *	213	0.8	

* The markers that were also differentially expressed between PC and non-cancer samples; other markers were only differentially expressed between PC and non-PC samples.

## Data Availability

The data in this study are openly available in the Gene Expression Omnibus accession number GSE85589 (www.ncbi.nlm.nih.gov/geo).

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
