# Peer review of "Identification of Circulating Serum miRNAs as Novel Biomarkers in Pancreatic Cancer Using a Penalized Algorithm"

_ijms, 2021, doi:10.3390/ijms22031007_

Round 1

Reviewer 1 Report

In this study, Lee et al investigated to identify non-invasive miRNA biomarkers and to build a prediction model for the diagnosis of PC. Overall, it is a well-designed study and the data interpretation and presentation is quite good. Addressing some of the following concerns should further strengthen the manuscript:   Comments 1- How many miRNAs common in your data with already reported in tissues biopsy analysis (see ref2, and line 48 in your paper)? 2- Out of 103 miRNA which is differentially express in PC VS Non-PC, how many are up and down regulated. Can you Include this in table-S1? (See line 205 in paper). Same apply for table S2. 3- Resolution of fig 2 is not good. Legend is not visible. 4- Out of 29 miRNA which you presented as final PC- specific biomarker. It is basically a biomarker for cancer not specific for PC cancer. You have also highlighted in fig 3. But in paper you have written this as a PC specific, please elaborate.  5- From your data its looks like only 11 PC specific miRNA you have reported. Can you please modify table 2 and highlight the 11 PC specific and 28 common miRNA in this table? 6- Finally, for validation you have selected 10 miRNA. What is the basis of selection? Does it is from 11 PC or it has 28 common miRNA too. 7- You mention that early biomarker detection is important. At which stage of PC your identified biomarker is reposted how it is different from previously reposted technique (serum levels of CA19-9) with respect to age of cancer (not sensitivity)? 8- Please explain little bit more about primary analysis about data QC. For example, how did you identify total 2,578 miRNA (Line 126 in paper). What is output of your miRNA data how much did you filter out during primary analysis? 9- Have you uploaded miRNA data on NCBI if yes please share that information in paper?  

Author Response

January 10, 2021

Dear Editors:

Thank you for your thoughtful consideration of our manuscript entitled “Identification of circulating serum miRNAs as novel biomarkers in pancreatic cancer using a penalized algorithm” by Jaehoon Lee et al. for publication in the International Journal of Molecular Sciences. The manuscript ID is IJMS-1053597.

We are thankful for your constructive comments, which have helped us to considerably improve and clarify the manuscript. We have responded to your comments in a point-by-point manner below. We hope that the changes incorporated into the revised manuscript satisfactorily address your concerns.

Thank you for your consideration. We hope our manuscript is now suitable for publication in your journal.

Sincerely,

Si Young Song, MD PhD

Department of Internal Medicine,

Yonsei University College of Medicine,

50-1 Yonsei-ro, Seodaemun-gu, Seoul 120-752, Korea

Phone: +82-2-2228-2684

FAX: +82-2227-7900

Editor Comments:

Reviewer 1

Comments and Suggestions for Authors

In this study, Lee et al investigated to identify non-invasive miRNA biomarkers and to build a prediction model for the diagnosis of PC. Overall, it is a well-designed study and the data interpretation and presentation is quite good. Addressing some of the following concerns should further strengthen the manuscript:  

Comments

1- How many miRNAs common in your data with already reported in tissues biopsy analysis (see ref2, and line 48 in your paper)?

We updated the following sentence & references.

Revised version Lines 320–321 (Discussion): ” Some of these (miR-155-5p, miR-939-5p, miR-346, miR-3064-5p, miR-661, miR-26a-5p, miR-200b-5p, and miR-218-5p) were found to be differentially expressed in PC tissues [16-32].

2- Out of 103 miRNA which is differentially express in PC VS Non-PC, how many are up and down regulated. Can you Include this in table-S1? (See line 205 in paper). Same apply for table S2.

We included that information by marking an asterisk (*) in Tables S1 and S2. Positive statistics denote upregulated miRNA in PC samples and negative statistics denotes downregulated miRNAs in PC samples.

3- Resolution of fig 2 is not good. Legend is not visible.

We revised Figure 1 & Figure 2 with a higher resolution to increase clarity of the image.

4- Out of 29 miRNA which you presented as final PC- specific biomarker. It is basically a biomarker for cancer not specific for PC cancer. You have also highlighted in fig 3. But in paper you have written this as a PC specific, please elaborate.

Thank you for raising your concerns. We identified 39 miRNAs as PC-specific biomarkers. We regarded 39 miRNAs as PC-specific biomarkers because they all could differentiate PC patients from non-PC. Non-PC includes patients with other cancers and healthy controls. Through each of those miRNAs or their combination, non-PC samples can be well diagnosed as non-PC with reasonable AUC in our study.

5- From your data its looks like only 11 PC specific miRNA you have reported. Can you please modify table 2 and highlight the 11 PC specific and 28 common miRNA in this table?

Please see the above answer. (It gave us a good opportunity to rethink of the meaning of “PC-specific” marker in the paper.) 39 miRNAs are regarded as PC-specific biomarkers. Among the 39 miRNAs, 28 miRNAs can also be used to differentiate PC patients from non-cancer controls, but 11 miRNAs cannot be used to differentiate PC patients from non-cancer controls. 28 miRNAs are marked with an asterisk (*) in Table 2.

6- Finally, for validation, you have selected 10 miRNA. What is the basis of selection? Does it is from 11 PC or it has 28 common miRNA too.

Thank you for mentioning this important point about our data. Due to the limited amount of serum samples and RNAs extracted from serum samples, we could only analyze 10 miRNAs from extracted RNAs per patient sample using the TaqMan Advanced miRNA Assays kit. Ten miRNAs were randomly selected to confirm the microarray analysis.

7- You mention that early biomarker detection is important. At which stage of PC your identified biomarker is reposted how it is different from previously reposted technique (serum levels of CA19-9) with respect to age of cancer (not sensitivity)?

We used PC patient samples in all stages, including stage 1 (n=4) and 2 (n=12) (Table 1), and the selected miRNAs were differentially expressed in all PC stages compared to normal samples. CA19-9 is the most commonly used PC biomarker, and CA19-9 expression is comparably low in early stage cancer, resulting in false negative detection in PC.

8- Please explain little bit more about primary analysis about data QC. For example, how did you identify total 2,578 miRNA (Line 126 in paper). What is output of your miRNA data how much did you filter out during primary analysis?

Thank you for your attention to detail. To increase the details about filtering in the manuscript, we added the following sentence:

Revised version Lines 121-126: “The Affymetrix GeneChip miRNA 4.0 Array provides 100% miRBase v20 coverage (www.mirbase.org) using a one-color approach. The Chip contains 6,658 human probe sets, which include pre-mature miRNAs (n=2025) and other small RNAs (n=1,996), including internal and negative controls. For further analysis, we extracted 2,578 mature human miRNAs, from all probe sets.”

We focused on mature miRNAs among all probe sets because mature miRNAs are produced from pre-mature miRNAs. One exception is that we did not conduct any further filtering process.

9- Have you uploaded miRNA data on NCBI if yes please share that information in paper?

In revised version Lines 410-411 (Data Availability Statement section), we have mentioned

“The data in this study are openly available in the Gene Expression Omnibus accession number GSE85589 (www.ncbi.nlm.nih.gov/geo).”

Reviewer 2 Report

In this study the authors developed a diagnostic model with selected circulating miRNAs (c-miRNAs) as the biomarkers for pancreatic cancer. They  investigated the samples from 63 patients with PC and 63 non-PC control patients for the identification of miRNA biomarkers, and an additional 25 PC samples and 81 ICC samples were used for the validation. They examined the sensitivity and specificity of their model through double cross-validation and obtained 0.93 as the mean sensitivity and 0.93 as the mean specificity compared with the common marker CA19-9 expression. For additional validation, the qRT-PCR results indicated that miR-155-5p, miR-4284, miR-346, miR-7145-5p, miR-5100, miR-661, miR-22-3p, miR-4486, let-7b-5p, and miR-4703-5p were differentially expressed in PC patient samples using additional independent 17 PC, 8 ICC, and 8 healthy control samples. This study is based on the bioinformatics analysis and qRT-PCR results and I have some concerns as following.

Major Comments:

  1. Several studies have highlighted the potential of c-miRNAs as biomarkers for the early diagnosis of cancer also including pancreatic cancer. Please mentioned the novelty of  this manuscript.
  2. What is the benefit or innovation of the diagnostic model which was established in this manuscript compared with the other methods? Because the significant miRNAs can be selected via different common bioinformatics analysis methods. The authors also mentioned “if we used only non-cancer samples as controls, the biomarkers led to many false positives... (line 211)”, please explain this point, otherwise this model would not be suitable.
  3. Please provided the complete data on significant c-miRNAs expression in all the samples from different groups (e.g heatmap, tables... )
  4. Please include the “analysis of statistics”, otherwise the data has no credibility, especially the qRT-PCR results.
  5. Please explain how to select 10 c-miRNAs from the 39 c-miRNAs Why didn’t choose the others (e.g. significance, function... )? Additionally, some of the markers are already  known.
  6. To further screen the biomarkers, the number of samples can be incensed and detect the c-miRNAs expression in patient tissue.
  7. Please revise the “materials and methods” part, especially 2.6, to match the format and explain in detail or cite literature because it is confused.
  8. Due to the low concentration of the c-miRNAs in the blood, ”Samples were centrifuged at 4 °C for 20 min at 3,000 × g.(line 94)” would not be enough for analysis. Can preform high speed centrifuge (e.g. 20 000 RCF for 10 minutes) after that  and  isolate exosomes from serum samples.
  9. Please revise the conclusion or title because of the unclear purpose of the study (model or biomarkers).
  10. Please check and update the references, some information is old (e.g.line 35).

Author Response

January 10, 2021

Dear Editors:

Thank you for your thoughtful consideration of our manuscript entitled “Identification of circulating serum miRNAs as novel biomarkers in pancreatic cancer using a penalized algorithm” by Jaehoon Lee et al. for publication in the International Journal of Molecular Sciences. The manuscript ID is IJMS-1053597.

We are thankful for your constructive comments, which have helped us to considerably improve and clarify the manuscript. We have responded to your comments in a point-by-point manner below. We hope that the changes incorporated into the revised manuscript satisfactorily address your concerns.

Thank you for your consideration. We hope our manuscript is now suitable for publication in your journal.

Sincerely,

Si Young Song, MD PhD

Department of Internal Medicine,

Yonsei University College of Medicine,

50-1 Yonsei-ro, Seodaemun-gu, Seoul 120-752, Korea

Phone: +82-2-2228-2684

FAX: +82-2227-7900

Editor Comments:

Reviewer 2

Comments and Suggestions for Authors

In this study the authors developed a diagnostic model with selected circulating miRNAs (c-miRNAs) as the biomarkers for pancreatic cancer. They investigated the samples from 63 patients with PC and 63 non-PC control patients for the identification of miRNA biomarkers, and an additional 25 PC samples and 81 ICC samples were used for the validation. They examined the sensitivity and specificity of their model through double cross-validation and obtained 0.93 as the mean sensitivity and 0.93 as the mean specificity compared with the common marker CA19-9 expression. For additional validation, the qRT-PCR results indicated that miR-155-5p, miR-4284, miR-346, miR-7145-5p, miR-5100, miR-661, miR-22-3p, miR-4486, let-7b-5p, and miR-4703-5p were differentially expressed in PC patient samples using additional independent 17 PC, 8 ICC, and 8 healthy control samples. This study is based on the bioinformatics analysis and qRT-PCR results and I have some concerns as following.

Major Comments:

  1. Several studies have highlighted the potential of c-miRNAs as biomarkers for the early diagnosis of cancer also including pancreatic cancer. Please mentioned the novelty of this manuscript.

Thank you for pointing out the novelty of this manuscript. We have highlighted the novelty of this manuscript in three ways and they are as follows:

1) In this study, training samples consisted of various types of cancer patients and healthy controls in order to uncover PC-specific biomarkers. 2) Through an appropriate penalized algorithm, we obtained very high sensitivity and specificity for PC-specific diagnosis. 3) We also obtained several novel biomarkers that have not yet been reported for PC-diagnosis.

We revised some sentences in the ‘conclusions’ section

Revised version Lines 340-342: “In this study, we identified 39 circulating miRNAs as PC-specific diagnostic markers using penalized methods. They include several novel biomarkers that have not yet been reported for PC-diagnosis.”

Revised version Lines 348-350:” Overall, while we are convinced that our identified miRNA biomarkers based on the PC-specific diagnosis model improve the detection rate for PC, further validation studies will be needed in the future.”

  1. What is the benefit or innovation of the diagnostic model which was established in this manuscript compared with the other methods? Because the significant miRNAs can be selected via different common bioinformatics analysis methods. The authors also mentioned “if we used only non-cancer samples as controls, the biomarkers led to many false positives... (line 211)”, please explain this point, otherwise this model would not be suitable.

Thank you for raising your concerns.

1) For the selection of reliable biomarkers from many miRNAs, we should consider 1) a combination of effects from multiple core miRNAs and 2) filtering negative effects caused by irrelevant miRNAs. In whole-genome or whole-sequencing studies that treat millions of markers, the penalized (=regularized) method is often used to solve these issues. We conducted a penalized SVM with SCAD to select multiple biomarkers from more than two thousand miRNAs. The SCAD penalty is well suited to identify the set of true signal features owing to the oracle property.

We have supplemented information on the benefit of our diagnostic model in the “discussion” section (Revised version Lines 305-307): “In order to consider joint effects from multiple core miRNAs and filter the negative effects caused by irrelevant miRNAs, we conducted a penalized SVM with SCAD penalty.”

2) We used non-PC samples (other cancer patients, non-cancer patients, or healthy people) as controls. If we did not include other cancer patients as controls, some other cancer patients could be diagnosed as PC because some cancers have similar mechanisms to pancreatic cancer.

We added a comment with examples for better understanding (Revised version Lines 211-213): “Thus, if we used only non-cancer samples as controls, the biomarkers led to many false positives (for example, the biomarker could diagnose some ICC patients as PC patients) and”

  1. Please provided the complete data on significant c-miRNAs expression in all the samples from different groups (e.g heatmap, tables... )

We added heatmaps in Figure S1 and S2 (Supplementary Materials).

  1. Please include the “analysis of statistics”, otherwise the data has no credibility, especially the qRT-PCR results.

We added the z-statistic column in Table S1, S2, and the statistical analysis in the qRT-PCR results in Figure 5.

  1. Please explain how to select 10 c-miRNAs from the 39 c-miRNAs candidates. Why didn’t choose the others (e.g. significance, function... )? Additionally, some of the markers are already known.

We analyzed miRNAs by RT-PCR to confirm the differential expression analyzed by the microarray analysis. Due to the limited amount of serum samples, we could only analyze 10 miRNAs from extracted RNAs per patient sample using TaqMan Advanced miRNA Assays kit, and miRNAs were randomly selected for confirmation.

  1. To further screen the biomarkers, the number of samples can be incensed and detect the c-miRNAs expression in patient tissue.

Unfortunately, we do not have the same patient’s tissues to analyze.

  1. Please revise the “materials and methods” part, especially 2.6, to match the format and explain in detail or cite literature because it is confused.

Thank you for your attention to detail.

We revised 2.6 and 2.7 overall to increase understanding and cited the references for double cross-validation.

For more details about filtering, we also added the following sentence.

Revised version Line 123-125: “This Chip contains 6,658 human probe sets, which include pre-mature miRNAs (n=2025) and other small RNAs (n=1,996), including internal and negative controls. For further analysis, we extracted 2,578 human mature miRNAs from all probe sets.”

  1. Due to the low concentration of the c-miRNAs in the blood, ”Samples were centrifuged at 4 °C for 20 min at 3,000 × g.(line 94)” would not be enough for analysis. Can preform high speed centrifuge (e.g. 20 000 RCF for 10 minutes) after that and isolate exosomes from serum samples.

Thank you for your comment. Blood samples collected in serum blood tubes were centrifuged at 4°C for 20 min at 3,000 × g to collect serum, and then we used a serum miRNA purification kit (phenol/chloroform-based) to isolate miRNAs.

  1. Please revise the conclusion or title because of the unclear purpose of the study (model or biomarkers).

Thank you for your advice. The title has been revised to “Identification of circulating serum miRNAs as novel biomarkers in pancreatic cancer using a penalized algorithm”

  1. Please check and update the references, some information is old (e.g.line 35).

We revised Lines 36-38 (revised version): ”According to cancer statistics in 2020, the 5-year survival rate of patients with PC is 9%, although that of patients with localized PC is higher than 37%, based on people diagnosed with pancreatic cancer between 2009 and 2015 [1].”

and updated the reference to the 2020 version.

For more relevant and current information, we added 9 additional references to the paper.
